# High-fidelity spin and optical control of single silicon-vacancy centres in silicon carbide

Roland Nagy[1], Matthias Niethammer[1], Matthias Widmann[1], Yu-Chen Chen[1], Péter Udvarhelyi[2,3], Cristian Bonato[4], Jawad Ul Hassan [5], Robin Karhu[5], Ivan G. Ivanov[5], Nguyen Tien Son[5], Jeronimo R. Maze[6,7], Takeshi Ohshima [8], Öney O. Soykal [9], Ádám Gali [2,10], Sang-Yun Lee [11], Florian Kaiser[1] & Jörg Wrachtrup[1]

Scalable quantum networking requires quantum systems with quantum processing capabilities. Solid state spin systems with reliable spin–optical interfaces are a leading hardware in this regard. However, available systems suffer from large electron–phonon interaction or fast spin dephasing. Here, we demonstrate that the negatively charged silicon-vacancy centre in silicon carbide is immune to both drawbacks. Thanks to its $^4A_2$ symmetry in ground and excited states, optical resonances are stable with near-Fourier-transform-limited linewidths, allowing exploitation of the spin selectivity of the optical transitions. In combination with millisecond-long spin coherence times originating from the high-purity crystal, we demonstrate high-fidelity optical initialization and coherent spin control, which we exploit to show coherent coupling to single nuclear spins with ~1 kHz resolution. The summary of our findings makes this defect a prime candidate for realising memory-assisted quantum network applications using semiconductor-based spin-to-photon interfaces and coherently coupled nuclear spins.

[1] 3rd Institute of Physics, University of Stuttgart and Institute for Quantum Science and Technology IQST, 70569 Stuttgart, Germany. [2] Institute for Solid State Physics and Optics, Wigner Research Centre for Physics, Hungarian Academy of Sciences, BudapestPO Box 49H-1525, Hungary. [3] Department of Biological Physics, Eötvös Loránd University, Pázmány Péter sétány 1/A, H-1117 Budapest, Hungary. [4] Institute of Photonics and Quantum Sciences, SUPA, Heriot-Watt University, Edinburgh EH14 4AS, UK. [5] Department of Physics, Chemistry and Biology, Linköping University, SE-58183 Linköping, Sweden. [6] Facultad de Física, Pontificia Universidad Católica de Chile, Santiago 7820436, Chile. [7] Research Center for Nanotechnology and Advanced Materials CIEN-UC, Pontificia Universidad Católica de Chile, Santiago 7820436, Chile. [8] National Institutes for Quantum and Radiological Science and Technology, Takasaki, Gunma 370-1292, Japan. [9] Naval Research Laboratory, Washington, DC 20375, USA. [10] Department of Atomic Physics, Budapest University of Technology and Economics, Budafoki út 8, H-1111 Budapest, Hungary. [11] Center for Quantum Information, Korea Institute of Science and Technology, Seoul 02792, Republic of Korea. Correspondence and requests for materials should be addressed to S.-Y.L. (email: sangyun.lee236@gmail.com) or to F.K. (email: f.kaiser@pi3.uni-stuttgart.de)

Optically addressable single spins in solids are a promising basis for establishing a scalable quantum information platform[1–3]. Electron spins are naturally suited for nanoscale quantum sensing[4–9], and for controlling nuclear spins that enable quantum information storage and computation[10–12]. Combined with an efficient spin-to-photon interface[13,14], this enables fast optical spin manipulation[13,15], entangling multiple quantum systems over long distances[16,17], and the realisation of quantum networks[1,18]. In this perspective, several pivotal landmark demonstrations have been achieved with the nitrogen-vacancy (NV) centre in diamond[13,16,18,19], and recently the divacancy centre in silicon carbide (SiC)[14]. However, implementation of scalable nano-photonics structures in the vicinity of those systems has proven to be detrimental for spin and optical stability and coherence[20,21]. New systems try to overcome those detrimental interactions through a high degree of symmetry. For example, the negatively charged silicon-vacancy centre in diamond shows high optical stability due to inversion symmetry[22,23]. However, the pronounced spin-phonon coupling necessitates millikelvin temperatures for decent spin coherence times[24]. Among colour centres in SiC, an efficient spin-to-photon interface has been demonstrated by divacancy spins in SiC[14], however, the zero-phonon line (ZPL) of divacancies is notoriously sensitive to stray electric fields similarly to that of NV centre in diamond[25].

So far, no known quantum system in solids demonstrated all the above specifications simultaneously. We suggest an alternative pathway to solving the above-mentioned issues, i.e., we resort to systems with highly symmetric ground and excited state symmetries and little redistribution of electron density between the two states[26]. The resulting little change in electric dipole moments between the two states effectively decouples the system from electric field fluctuations. Strong optical transitions can still be obtained, provided that ground and excited states wavefunctions have alternating phases. In previous works, we demonstrated that the $V_{Si}$ shows low geometrical relaxation under excitation, leading to very high emission into the zero phonon line[27]. Additionally, we performed preliminary investigations on coherent spin control and optical readout with limited contrast due to the lack of high-fidelity spin initialisation and readout procedures[27,28].

In this report, we show that the hexagonal lattice site silicon vacancy ($V_{Si}$) in the 4H polytype SiC close to ideally matches all criteria. Compared to our previous studies, the robust properties of $V_{Si}$ in the improved host crystal material allows us to observe for the first time narrow and separated optical transitions of a single $V_{Si}$, which we address to demonstrate a high-fidelity spin-optical interface, suitable for quantum networking applications. Our findings prove that good quantum systems do not necessarily require inversion symmetry, which opens the door for new classes of quantum systems in numerous semiconductors and insulators.

## Results

### Silicon vacancies in silicon carbide
The structure of the 4H-SiC crystal results in two non-equivalent sites for a $V_{Si}$. As shown in Fig. 1a, we investigate the defect centre that is formed by a missing silicon atom at a hexagonal lattice site and negatively charged ($V_1$ centre)[29]. The ground state has a spin quartet manifold ($S = 3/2$) with weak spin-orbit coupling, leading to millisecond spin relaxation time[27]. To study the centre's intrinsic optical and electron spin properties, we use a nuclear spin free isotopically purified 4H-SiC layer ($^{28}Si > 99.85\%$, $^{12}C > 99.98\%$), in which we create single centres by electron irradiation (see Methods). The defects are optically addressed by confocal microscopy at cryogenic temperatures ($T \sim 4$ K). We employ a 730 nm laser diode for off-resonant excitation. A wavelength-

tunable diode laser (Toptica DL pro) performs resonant excitation at 861 nm in the ZPL of the lowest optical transition with $A_2$ symmetry, known as V1 line. Another energetically higher transition to an electronic state of E symmetry, called V1' line[27,29,30], is not investigated in this work. Fluorescence emission is detected in the red-shifted phonon side band (875–890 nm). In addition, ground state spin manipulation is performed with microwaves (MW) that are applied via a 20 μm diameter copper wire. See Supplementary Note 1 for single photon emission characteristics of the identified single silicon vacancies.

### Excited state spectroscopy
Figure 1b illustrates the defect's energy level structure. At zero external magnetic field ($B_0 = 0$ G), ground and excited state manifolds show pairwise degenerate spin levels $m_S = \pm 1/2$ and $m_S = \pm 3/2$, with zero-field splittings (ZFS) of $2 \cdot D_{gs}$ and $2 \cdot D_{es}$, respectively. Previous studies constrained $2 \cdot D_{gs} < 10$ MHz[27,29,31] and here we determine $2 \cdot D_{gs} = 4.5 \pm 0.1$ MHz (see Methods). In order to investigate the excited state structure, we use resonant optical excitation. We apply a strong MW field at ~4.5 MHz to continuously mix the ground state spin population, and wavelength-tune simultaneously the 861 nm laser across the optically allowed transitions. As shown in Fig. 1c, we observe two strong fluorescence peaks, labelled $A_1$ and $A_2$. The peak separation of $980 \pm 10$ MHz corresponds to the difference between the ground and excited state ZFS. As shown in the Supplementary Note 2, we use coherent spin manipulation to infer a positive excited state ZFS, i.e., $2 \cdot D_{es} = 985 \pm 10$ MHz, which is in line with the results of first-principles density functional theory. To determine optical selection rules, we apply an external magnetic field of $B_0 = 92$ G precisely aligned along the uniaxial symmetry axis of $D_{gs}$, which is parallel to the c-axis of 4H-SiC, such that the ground state Zeeman spin-splitting exceeds the optical excitation linewidth (see Fig. 1b). We observe no shift of the optical resonance lines, corroborating our assignment of $A_{1,2}$ as spin-conserving optical transitions between ground and excited states. Further, we observe no spin-flip transitions, which would show up as additional peaks in the spectra in Fig. 1c at approximately ±258 MHz (see Supplementary Note 3). However, as we will show later, spin-flips can still occur through non-radiative decay channels. Moreover, as the presence of a magnetic field does not alter the peak separation, we confirm that ground and excited state g-factors are identical (see Supplementary Note 3)[32]. Figure 1d shows repetitively recorded excitation spectra for which we find exceptionally stable lines over an hour time-scale.

### Stable optical resonance
To underline that the defect's wavefunction symmetry indeed largely decouples its optical transition energy from stray charges in its local environment, we perform resonant excitation studies on four other defects. As shown in Fig. 1f, the peak separation of all defects is nearly identical within 19 MHz. In addition, all resonant absorption lines are inhomogeneously distributed over only a few hundred MHz, allowing us to identify several defects with overlapping emission. The observed inhomogeneity is likely due to local strain near the surface of the solid immersion lens which is fabricated to improve the photon collection efficiency. We also can observe small additional peaks in a few defects, whose origin is not yet investigated.

Advanced quantum information applications based on spin-photon entanglement require that the quantum system emits transform-limited photons. We measure this via the excitation linewidth of the optical transitions. In Fig. 1e, we show that for excitation intensities below 1 W/cm², the linewidth approaches 60 MHz. Considering the 5.5 ns excited state lifetime[27], this is

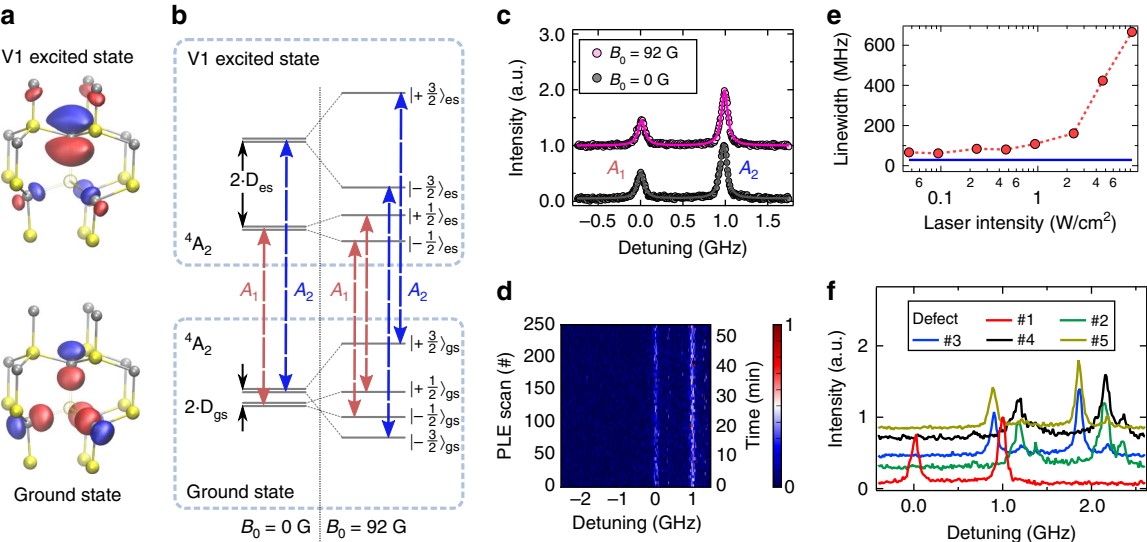

**Fig. 1** Optical transitions of the silicon vacancy in 4H-SiC. **a** Crystalline structure of 4H-SiC with a silicon vacancy centre at a hexagonal lattice site. Upper (lower) figure shows the square moduli of the defect wave functions of the V1 excited (ground) state, as calculated by density functional theory. Red (blue) shaded areas symbolise that the wave function has a positive (negative) sign. The yellow and grey spheres represent silicon and carbon atoms, respectively, and the crystallographic c-axis is aligned vertically in this figure. **b** Ground and excited state level scheme with and without a magnetic field applied along the c-axis. Red (blue) optical transitions labelled $A_1$ ($A_2$) connect spin levels $m_S = \pm 1/2$ ($m_S = \pm 3/2$). **c** Resonant absorption spectrum of a single vacancy centre at $B_0 = 0$ G and $B_0 = 92$ G. Lines are fits using a Lorentzian function. **d** Repetitive resonant absorption scans at $B_0 = 92$ G over 52 min without any sign of line wandering. The colour bar indicates the normalized intensity in arbitrary units. **e** Absorption linewidth of the peak $A_2$ as a function of the resonant pump laser intensity. Below 1 W/cm² no power broadening is observed and the linewidth is close to transform limited as indicated by the blue line. **f** Resonant absorption spectra of five single defect centres, showing several defects with almost identical separation between two ZPLs and linewidth but inhomogeneously distributed

only twice the Fourier-transform limit, and might be explained by small residual spectral diffusion. For example, previous studies on NV centres in diamond and divacancies in SiC emphasize the impact of the sample quality, especially the impurity concentration, on the linewidth of the resonant optical transitions[14,33]. We indeed find that a higher defect concentration leads to broader linewidths (see Supplementary Note 4). This suggests that low-damage defect generation techniques and Fermi level tuning via adapted doping can further enhance the quality of our optical transitions. We also measure the Stark shift tuning coefficient, which is at least one order of magnitude smaller compared to NV centres in diamond (see Supplementary Note 5). As explained in the Supplementary Note 5, this excellent optical stability is intimately linked to the wavefunction symmetry of the $V_{Si}$. Interestingly, unlike diamond, SiC cannot host defects with inversion symmetry due to non-centrosymmetry. As a consequence, the $V_{Si}$ shows non-zero ground and excited state dipole moments, which directly couple to electric fields. However, for the $V_{Si}$, those dipole moments are nearly identical, such that the optical transition energy is not altered by electric fields. Note that this does not preclude the existence of a strong dipole transition between ground and excited states, but it restricts the orientation to the symmetry axis of the defect, which is the c-axis of the crystal[26].

**Efficient control of the electron spin states.** Realising a spin-to-photon interface for quantum information applications requires high-fidelity spin state initialisation, manipulation and readout[34]. Previous theoretical models[35] and ensemble-based measurements[27] have indicated that continuous off-resonant optical excitation of $V_{Si}$ eventually leads to a decay into a metastable state manifold (MS), followed by rather less-selective relaxation into the ground state spin manifold. Here, we use resonant optical excitation to strongly improve spin state selectivity. As shown in

Fig. 2a, we apply a magnetic field of $B_0 = 92$ G, allowing us to selectively address transitions within the ground state spin manifold via MW excitation. We first excite the system along the $A_2$ transition (linking the $m_S = \pm 3/2$ spin states), which eventually populates the system into the $m_S = \pm 1/2$ ground state via decays through the MS (see Supplementary Note 6). Then, we perform optically detected magnetic resonance (ODMR). For this, we apply narrowband microwave pulses in the range of 245–75 MHz, followed by fluorescence detection during an optical readout pulse on the $A_2$ transition. As shown in Fig. 2b, we observe two spin resonances at the magnetic dipole allowed transitions, $\left|-\frac{1}{2}\right\rangle_{gs} \leftrightarrow \left|-\frac{3}{2}\right\rangle_{gs}$ and $\left|+\frac{1}{2}\right\rangle_{gs} \leftrightarrow \left|+\frac{3}{2}\right\rangle_{gs}$ at 253.5 MHz (MW$_1$) and 262.5 MHz (MW$_3$), respectively. To observe the centre resonance at 258.0 MHz (MW$_2$), we need to imbalance the population between $\left|-\frac{1}{2}\right\rangle_{gs}$ and $\left|+\frac{1}{2}\right\rangle_{gs}$. To this end, we combine the initial resonant laser excitation pulse on $A_2$ with MW$_3$. MW$_3$ continuously depopulates $\left|+\frac{1}{2}\right\rangle_{gs}$, such that the system is eventually pumped into $\left|-\frac{1}{2}\right\rangle_{gs}$. After applying microwaves in the range between 245–275 MHz, the optical readout pulse is also accompanied by MW$_3$, which transfers population from $\left|+\frac{1}{2}\right\rangle_{gs}$ to $\left|+\frac{3}{2}\right\rangle_{gs}$. This effectively allows us to detect a fluorescence signal from spin population in $\left|+\frac{1}{2}\right\rangle_{gs}$. As shown in Fig. 2c, we observe the two expected resonances for $\left|-\frac{1}{2}\right\rangle_{gs} \leftrightarrow \left|-\frac{3}{2}\right\rangle_{gs}$ and $\left|-\frac{1}{2}\right\rangle_{gs} \leftrightarrow \left|+\frac{1}{2}\right\rangle_{gs}$ at 253.5 MHz (MW$_1$) and 258.0 MHz (MW$_2$), respectively. Unless mentioned otherwise, we now always initialise the system into $\left|-\frac{1}{2}\right\rangle_{gs}$ using the above-described procedure and read out the spin state population of the $\left|\pm\frac{3}{2}\right\rangle_{gs}$ levels with a 1 µs long laser pulse resonant with the $A_2$ transition.

We first demonstrate coherent spin state control via Rabi oscillations. Figure 3a shows the sequence for Rabi oscillations

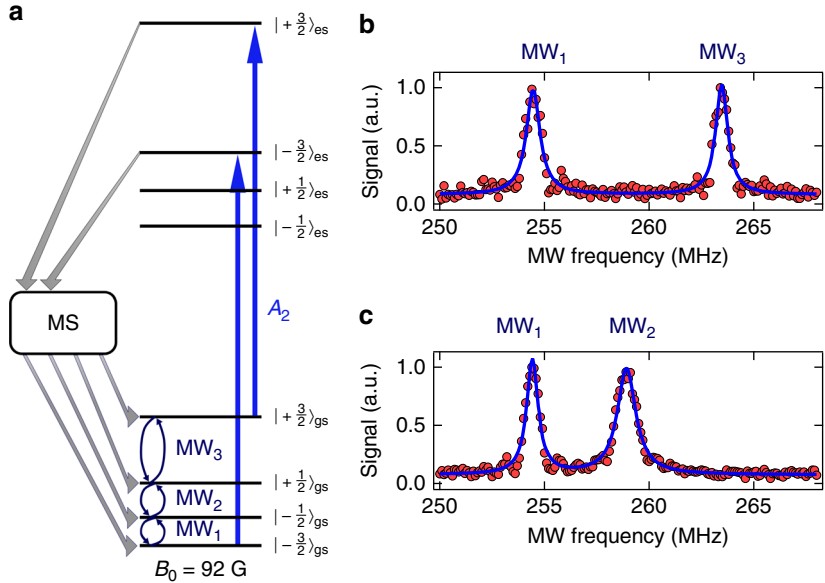

**Fig. 2** Optically detected magnetic resonance. **a** Level scheme indicating the used optical transition ($A_2$) and microwave fields $MW_1$, $MW_2$ and $MW_3$. Spin flips occur via nonradiative channels involving metastable states (MS). **b** ODMR signal of the ground state after initialising the system into $\left|\pm\frac{1}{2}\right\rangle_{gs}$. **c** ODMR signal after initialisation into $\left|-\frac{1}{2}\right\rangle_{gs}$. Blue lines are fits using Lorentzian functions. All data are normalised raw data, i.e. without background subtraction

between $\left|-\frac{1}{2}\right\rangle_{gs} \leftrightarrow \left|-\frac{3}{2}\right\rangle_{gs}$. After spin state initialisation, we apply $MW_1$ for a time $\tau_{Rabi}$ to drive population towards $\left|-\frac{3}{2}\right\rangle_{gs}$, followed by optical readout. To observe oscillations between $\left|-\frac{1}{2}\right\rangle_{gs} \leftrightarrow \left|+\frac{1}{2}\right\rangle_{gs}$, the system is initialised, $MW_2$ is applied for a time $\tau_{Rabi}$, followed by a population transfer from $\left|+\frac{1}{2}\right\rangle_{gs} \rightarrow \left|+\frac{3}{2}\right\rangle_{gs}$ using a $\pi$-pulse at $MW_3$, and eventual state readout. Figure 3b shows the experimental results from which we deduce Rabi oscillation frequencies of 257.5 kHz and 293.8 kHz. The frequency ratio of $1.14 \approx \sqrt{4/3}$ is in excellent agreement with the theoretical expectation for a quartet spin system[36]. In a next step, we proceed to measuring spin coherence times between the ground state levels $\left|-\frac{1}{2}\right\rangle_{gs}$ and $\left|-\frac{3}{2}\right\rangle_{gs}$. To this end, we perform a free induction decay (FID) measurement in which we replace the Rabi pulse in Fig. 3a by two $\frac{\pi}{2}$-pulses separated by a waiting time $\tau_{FID}$. The experimental data in Fig. 3c show a dephasing time of $T_2^* = 30 \pm 2\,\mu s$, which is comparable with state-of-the-art results reported for NV centres in isotopically ultrapure diamonds[37]. To measure the spin coherence time $T_2$, we use a Hahn-echo sequence by adding a refocusing $\pi$-pulse in the middle of two $\frac{\pi}{2}$-pulses. As shown in Fig. 3d, we measure a spin coherence time of $T_2 = 0.85 \pm 0.12$ ms, which is longer than $T_2$ of a single $V_2$ centre, a $V_{Si}$ at another non-equivalent lattice site, in a commercially available SiC wafer measured at room temperature[28]. This improvement is thanks to the lower impurity concentration in the used CVD grown SiC layer (see Methods and Supplementary Note 7). For an isotopically purified system, like the one used in the present experiment, however, we expect somewhat longer coherence times. As outlined in the Supplementary Note 7, we attribute the origin to be nearby defect clusters created by microscopic cracks induced by the CVD wafer cutting process using a dicing saw. Hence, optimised sample growth, cutting and annealing should increase dephasing times to several tens of milliseconds[38]. Unambiguous identification of the decoherence source will require systematic investigation by controlling spin bath dynamics[39,40].

**Coherently coupled nuclear spins**. Quite interestingly, the initial part of the spin echo decay of this particular defect shows pronounced oscillations resulting from the hyperfine coupling of the electron spin to nuclear spins. Being initially polarized along the $z$-axis in the Bloch sphere, and with microwave pulses polarized along the $x$-axis, the echo sequence measures[41]

$$-S_y = 1 - \frac{1}{k}\left[2 - 2\cos\left(\omega_\alpha \tau\right) - 2\cos\left(\omega_\beta \tau\right) + \cos\left(\omega_- \tau\right) + \cos\left(\omega_+ \tau\right)\right].$$

(1)

Here, $k$ is the modulation depth parameter $k = \left(\frac{2\omega_\alpha \omega_\beta}{A_\perp \omega_I}\right)^2$, with $\omega_I$ being the nuclear Larmor frequency, and $\omega_{\alpha,\beta} = \left[\left(\omega_I + m_{\alpha,\beta}A_\parallel\right)^2 + \left(m_{\alpha,\beta}A_\perp\right)^2\right]^{\frac{1}{2}}$. Here, $m_\alpha = -\frac{3}{2}$ and $m_\beta = -\frac{1}{2}$ are the spin projections of the involved ground states with respect to the $z$-axis. Further, $\omega_\pm = \omega_\alpha \pm \omega_\beta$, and $A_{\perp,\parallel}$ are the orthogonal and parallel hyperfine components, respectively. Essentially, $S_y$ is modulated because of quantum beats between hyperfine levels, which not only show the nuclear frequencies $\omega_\alpha$, but also their sum and difference frequencies $\omega_\pm$. The Fourier transformation reveals strong frequency components at $\omega_\alpha/2\pi = 77.9 \pm 0.1$ kHz and $\omega_\beta/2\pi = 76.0 \pm 0.1$ kHz. This small difference (1.9 kHz) is resolved thanks to the high sensitivity of the quantum probe, the electronic spin with a nearly millisecond-long coherence time. As $\omega_+$ is quite close to twice the Larmor frequency of a $^{29}Si$ nuclear spin ($\omega_I/2\pi \approx 77.9$ kHz), we conclude that the parallel hyperfine coupling $A_\parallel$ is weak compared to the Larmor frequency. In addition, from the inferred modulation depth parameter ($k = 0.15 \pm 0.02$), we infer that there is a sizeable difference between $A_\perp$ and $A_\parallel$. Indeed, from the fit to the data, we infer a purely dipolar coupling with strengths of $A_\perp \approx 29$ kHz and $A_\parallel \approx 10$ kHz, respectively. This allows inferring the relative position of the nuclear spin, using $A_\parallel = \frac{\eta_{Si}}{r^3}(3\cos^2\theta - 1)$ and $A_\perp = \frac{\eta_{Si}}{r^3}(3\sin\theta\cos\theta)$, in which $\eta_{Si} = 15.72$ MHz · Å$^3$ is the dipole-dipole interaction coefficient[42], $r$ is the distance between electron and nuclear spin, and $\theta$ is the polar angle between the

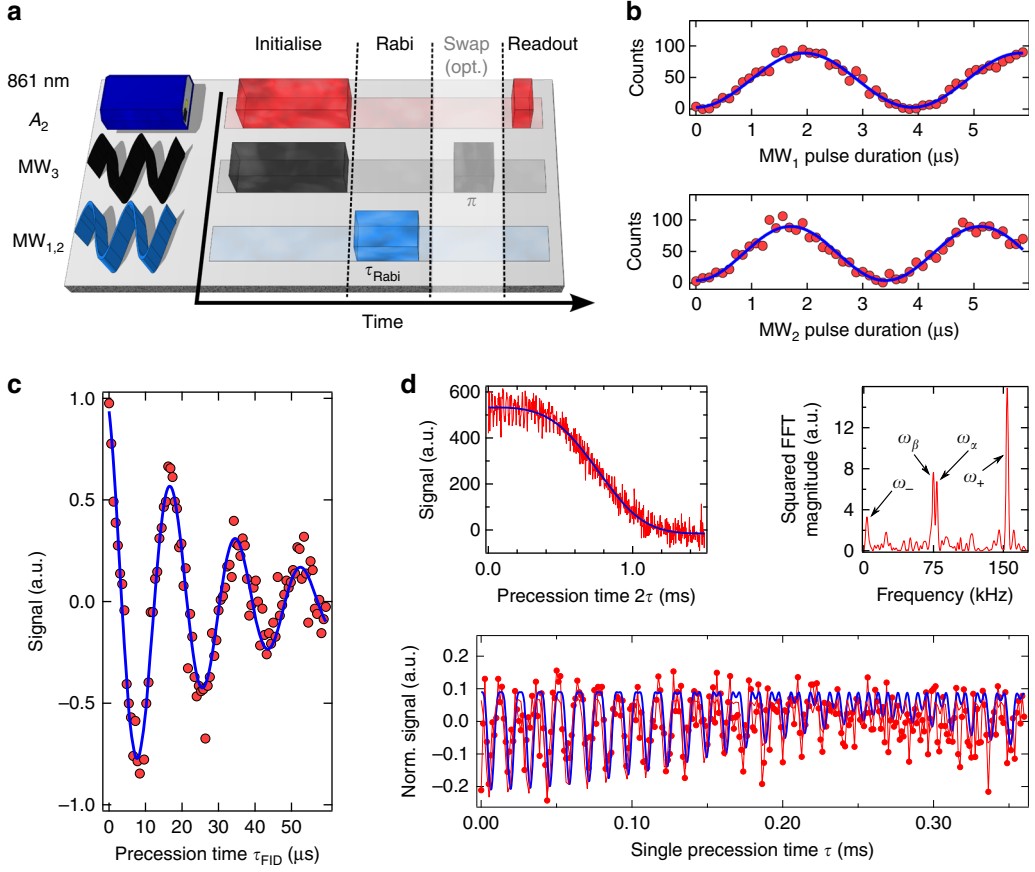

**Fig. 3** Spin manipulation and coherence. **a** Experimental sequence for observing Rabi oscillations. The system is always initialised into $\left|-\frac{1}{2}\right\rangle_{gs}$ using resonant excitation along $A_2$ and $MW_3$. This step is followed by a Rabi sequence ($MW_{1,2}$), an optional population swap $\left(\left|+\frac{1}{2}\right\rangle_{gs}\leftrightarrow\left|+\frac{3}{2}\right\rangle_{gs}\right)$, and optical readout. **b** Rabi oscillations for $\left|-\frac{1}{2}\right\rangle_{gs}\leftrightarrow\left|-\frac{3}{2}\right\rangle_{gs}$ (upper panel) and $\left|-\frac{1}{2}\right\rangle_{gs}\leftrightarrow\left|+\frac{1}{2}\right\rangle_{gs}$ (lower panel). Blue lines are sinusoidal fits. All data are raw data. The maximum contrast, $1-I_{max}/I_{min}$ is $97\pm1\%$. **c** Free induction decay measurement yielding $T_2^*=30\pm2\,\mu s$, and the blue line is a fit. **d** Hahn echo measurement and nuclear spin coupling. From the top left graph, we infer $T_2=0.8\pm0.12$ ms. Red lines are data and the blue line is a fit using a higher-order exponential function. The bottom panel is a zoom into the first part of the Hahn echo after subtraction of the exponential decay function and normalisation. Pronounced oscillations are observed, witnessing coherent coupling to a nearby nuclear spin. Data (red dots connected by lines) are fitted using Eq. (1) (blue line). The top right panel is a Fourier analysis of the normalised Hahn echo, showing four distinct frequency components through which a weakly coupled $^{29}$Si nuclear spin is identified

c-axis and the vector connecting the positions of the nuclear spin and electron spin. We obtain $r\approx11.6$ Å and $\theta\approx61°$. We emphasize that we did not observe hyperfine coupling in Hahn echo measurements on the other defects, however, an adapted crystal growth with nuclear spins at a percentage concentration should allow to identify multi-spin clusters that could be used as small-scale quantum computers[43].

**High fidelity optical initialization.** Besides excellent coherence times, a crucial requirement for quantum information applications is high-fidelity quantum state initialisation[34]. To optimise the spin state initialisation procedure, we vary the time interval $\tau_{init}$ of the initialising laser ($A_2$) and microwave fields ($MW_3$) and extract the populations in the four ground states via the contrast of Rabi oscillation measurements (for more details, see Methods). Figure 4a shows the experimental sequence. We first equilibrate all four ground state populations to within <1% with a 40 μs long off-resonant laser pulse[27]. Then, we apply the selective spin state initialisation procedure for a time $\tau_{init}$, followed by different Rabi sequences, and eventual state readout along the $A_2$ optical transition. Figure 4b shows the development of the extracted ground state populations for $\tau_{init}$ ranging from 0 to 80 μs. We achieve

initialisation fidelities up to $97.5\pm2.0\%$, which is comparable to previous demonstrations with colour centres in diamond[37] and mark the state-of-art value for colour centres in SiC.

## Discussion

The silicon vacancy centre in 4H-SiC satisfies a number of key requirements for an excellent solid-state quantum spintronics system. The high spectral stability and close to transform limited photon emission as well as large Debye Waller factor (>0.4)[27] promise good spin-photon entanglement generation rates. We also mention that almost no defect ionisation has been observed, except at high optical excitation powers beyond saturation. This is in stark contrast to NV centres in diamond where charge state verification is required before each experimental sequence[18]. By removing this requirement, spin-photon entanglement rates can be further speed up. The optical resonance excitation allows for high-fidelity spin-selective initialisation which result in 97% optical contrast of spin readout. The emission wavelength of 861 nm is favourable for fibre based long distance communication as ultralow-noise frequency converters to the telecom range already exist[44,45]. The measured overall emission rates in the phonon sideband are currently about $20\times10^3$ counts s$^{-1}$. Given the

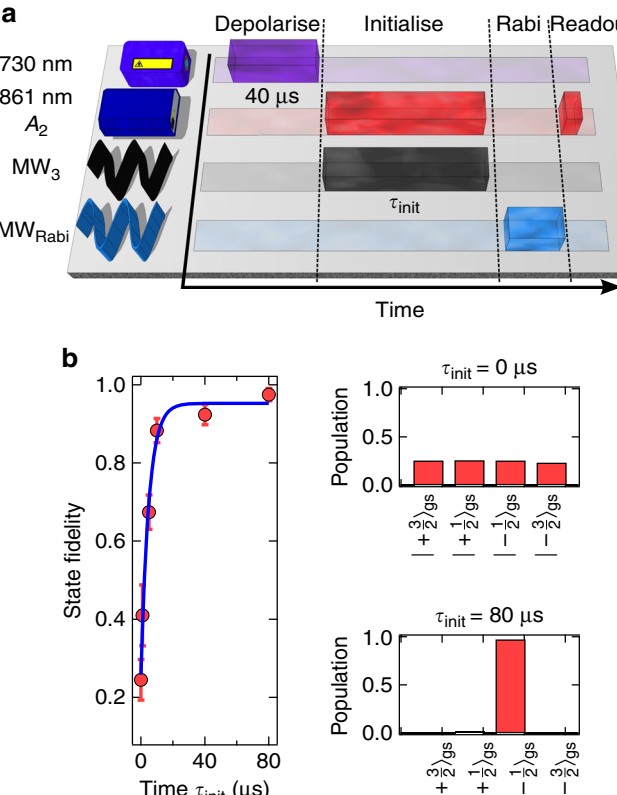

**Fig. 4** Electron spin initialisation fidelity. **a** Experimental sequence. Before each round, the ground state spin is depolarised using off-resonant excitation for 40 μs. Then, the system is initialised into $\left|-\frac{1}{2}\right\rangle_{gs}$. Ground state populations are inferred from Rabi oscillations and resonant optical readout. **b** Left side: Spin population in $\left|-\frac{1}{2}\right\rangle_{gs}$ as a function of the duration of the initialisation procedure. Up to 97.5% are achieved. The blue line is a fit using an exponential function. The error bars are estimated from the uncertainties in determining populations of the ground spin states, assuming that the main source of error is shot noise. Right side: Inferred spin populations in the four ground state sublevels without initialisation (top) and after 80 μs initialisation time

limited system detection efficiency of around 0.1%, due to the use of non-optimised optics and detectors, we estimate an overall photon emission rate of $\approx 10^{7} \, \mathrm{s}^{-1}$, limited by non-radiative intersystem crossings. Considering that a large fraction of the photons are emitted in the zero phonon line (>40%), this implies that spin-photon entanglement can be generated at rates of several tens to hundreds of Hz[18], currently mainly limited by the non-optimised light collection efficiency. Implementation into optical resonators[46,47] could further increase emission rates by reducing the excited state lifetime, and achieving a Purcell factor of 10–20 would already yield sufficient photon rates to accomplish quantum non-demolition readout of the spin state. The defect's very low strain coupling makes this strategy very promising. We showed also that spin coherence times, as well as readout and initialization fidelities are comparable with other systems, e.g., colour centres in diamond. This allowed us to observe hyperfine coupling, which promises access to long-lived quantum memories and quantum registers. The high optical contrast of ground spin state readout will also allow for single-shot readout of coupled nuclear spins[48]. In this perspective, the relatively small ground state ZFS, which is on the order of the typical hyperfine interaction, may prove to be beneficial for nuclear spin polarisation techniques, potentially leading to applications in high-contrast magnetic resonance imaging[49,50].

Additionally, excitation to the second excited state[27,30,35] is predicted to show significantly reduced intersystem crossing rates, thus further enhancing optical spin manipulation capabilities and single-shot readout of electronic spins, similar to protocols based on cold atoms or ions[51]. The silicon vacancy centre in 4H-SiC therefore has a great potential to become a leading contender in a wealth of quantum information applications.

## Methods

**Sample preparation**. The starting material is a 4H-$^{28}$Si$^{12}$C silicon carbide layer grown by chemical vapour deposition (CVD) on a n-type (0001) 4H-SiC substrate. The CVD layer is ~110 μm thick. The isotope purity is estimated to be [$^{28}$S]i > 99.85% and [$^{12}$C] > 99.98%, which was confirmed by secondary ion mass spectroscopy (SIMS) for one of the wafers in the series. After chemical mechanical polishing (CMP) of the top layer, the substrate was removed by mechanical polishing and the final isotopically enriched free-standing layer had a thickness of ~100 μm. Current-voltage measurements at room temperatures show that the layer is n-type with a free carrier concentration of ~6 × 10$^{13}$ cm$^{-3}$. This value is close to the concentration of shallow nitrogen donors of ~3.5 × 10$^{13}$ cm$^{-3}$, which was determined by photoluminescence at low temperatures. Deep level transient spectroscopy measurements show that the dominant electron trap in the layer is related to the carbon vacancy with a concentration in the mid 10$^{12}$ cm$^{-3}$ range. Minority carrier lifetime mapping of the carrier shows a homogeneous carrier lifetime of ~0.6 μs. Since the lifetime was measured by an optical method with high injection, the real lifetime is expected to be double, i.e., ~1.2 μs[52]. Such a high minority carrier lifetime indicates that the density of all electron traps should not be more than mid 10$^{13}$ cm$^{-3}$[53]. To generate a low density of silicon vacancy centres, we used room temperature electron beam irradiation at 2 MeV with a fluence of 10$^{12}$ cm$^{-2}$. The irradiation creates also carbon vacancies, interstitials, anti-sites and their associated defects, but their concentrations are expected to be below mid 10$^{12}$ cm$^{-3}$. After irradiation, the sample was annealed at 300 °C for 30 min to remove some interstitial-related defects. In order to improve light extraction efficiency out of this high refractive index material ($n \approx 2.6$), we fabricate solid immersion lenses using a focused ion beam milling machine (Helios NanoLab 650). The sample was cleaned for two hours in peroxymonosulfuric acid to remove surface contaminations[28].

**Experimental setup**. All the experiments were performed at a cryogenic temperature of 4 K in a Montana Instruments Cryostation. A home-built confocal microscope[27] was used for optical excitation and subsequent fluorescence detection of single silicon vacancies. Off-resonant optical excitation of single silicon vacancy centres was performed with a 730 nm diode laser. For resonant optical excitation at 861.4 nm towards V1 excited state we used an external cavity tunable diode laser (Toptica DL pro). The used laser power for the resonant excitation was adjustable from 0.5 to 500 nW. Fluorescence was collected in the red-shifted phonon sideband (875–890 nm) for which we used a tunable long-pass filter (Versa Chrome Edge from Semrock). To detect light, we used a near infrared enhanced single-photon avalanche photodiode (Excelitas). The used 4H-SiC sample was flipped to the side, i.e., by 90° compared to the c-axis, such that the polarisation of the excitation laser was parallel to the c-axis (E||c) which allows to excite the V1 excited state with maximum efficiency[27,30]. Note that the solid immersion lenses (SILs) were fabricated on this side surface. In order to manipulate ground state spin populations, microwaves are applied through a 20 μm thick copper wire located in close proximity to the investigated V1 defect centres.

**Inferring ground state spin initialisation fidelity**. Near deterministic ground state spin initialization has been demonstrated via optically pumping assisted by microwave spin manipulation (see Fig. 4a). We first apply the off-resonant laser excitation (730 nm) to prepare non-initialized spin states. We then initialize the system into $\left|-\frac{1}{2}\right\rangle_{gs}$ by resonant optical excitation along the $A_2$ transition, accompanied by microwaves resonant with the transition $\left|+\frac{1}{2}\right\rangle_{gs} \leftrightarrow \left|+\frac{3}{2}\right\rangle_{gs}$ (MW$_3$). To determine the population in each ground state spin level, we perform Rabi oscillations for the three allowed transitions, linking the levels $\left|-\frac{1}{2}\right\rangle_{gs} \leftrightarrow \left|-\frac{3}{2}\right\rangle_{gs}$ (MW$_1$), $\left|-\frac{1}{2}\right\rangle_{gs} \leftrightarrow \left|+\frac{1}{2}\right\rangle_{gs}$ (MW$_2$), and $\left|+\frac{1}{2}\right\rangle_{gs} \leftrightarrow \left|+\frac{3}{2}\right\rangle_{gs}$ (MW$_3$). Then, we read out the spin population in $\left|\pm\frac{3}{2}\right\rangle_{gs}$ by resonant excitation along the $A_2$ transition for 150 ns. Within such a short readout time, we can safely assume that the obtained fluorescence signal is proportional to the population in $\left|\pm\frac{3}{2}\right\rangle_{gs}$. Note that in order to obtain a signal from Rabi oscillations $\left|-\frac{1}{2}\right\rangle_{gs} \leftrightarrow \left|+\frac{1}{2}\right\rangle_{gs}$, an additional population swap (π-pulse at MW$_3$) between $\left|+\frac{1}{2}\right\rangle_{gs}$ and $\left|+\frac{3}{2}\right\rangle_{gs}$ is applied before state readout (see also Fig. 3a). We denote now the populations in all four ground states by $p_i$ where $i = \left\{-\frac{3}{2}, -\frac{1}{2}, +\frac{1}{2}, +\frac{3}{2}\right\}$ stands for the spin quantum number of each state. We then measure the fringe visibility of the obtained Rabi oscillation signal in order to infer ground state spin populations. The fringe visibility for Rabi oscillations between sublevels $\left|i\right\rangle_{gs}$ and $\left|j\right\rangle_{gs}$ with

$i, j = \left\{-\frac{3}{2}, -\frac{1}{2}, +\frac{1}{2}, +\frac{3}{2}\right\}$ and $|i - j| = 1$ is defined as,

$$\nu_{i,j} = \frac{I_{\max} - I_{\min}}{I_{\max} + I_{\min}}, \tag{2}$$

where $I_{\max}$ ($I_{\min}$) denotes the maximum (minimum) signal during the Rabi oscillation. Considering that our state readout is only sensitive to spin population in $\left|\pm\frac{3}{2}\right\rangle_{\mathrm{gs}}$, the three conducted Rabi oscillation experiments lead to the following fringe visibilities:

$$\nu_{3/2,1/2} = \left(p_{1/2} - p_{3/2}\right)/\left(2 \cdot p_{-3/2} + p_{1/2} + p_{3/2}\right) \tag{3}$$

$$\nu_{1/2,-1/2} = \left(p_{-1/2} - p_{1/2}\right)/\left(2 \cdot p_{-3/2} + p_{-1/2} + p_{1/2}\right) \tag{4}$$

$$\nu_{-1/2,-3/2} = \left(p_{-1/2} - p_{-3/2}\right)/\left(2 \cdot p_{3/2} + p_{-3/2} + p_{-1/2}\right). \tag{5}$$

In addition, the total population of all the ground states must be sum up to unity, i.e.,

$$p_{-3/2} + p_{-1/2} + p_{+1/2} + p_{+3/2} = 1. \tag{6}$$

As all observed Rabi oscillations start with a local minimum in fluorescence intensity, we can further assume that after initialization, the ground state populations fulfil:

$$p_{-3/2} \sim p_{+3/2} < p_{+1/2} < p_{-1/2}. \tag{7}$$

Solving the system of four equations under these constraints allows us to extract all four values of $p_i$, which is shown for the initialization times 0 μs and 80 μs in Fig. 4b.

**Magnetic field alignment**. In order to allow for selective ground state spin manipulation, level degeneracy has to be lifted. We do so by applying a magnetic field that has to be precisely aligned along the z-axis of the spin system in order to avoid the appearance of new mixed eigenstates. The z-axis of the spin system is parallel to the c-axis of the 4H-SiC crystal[31]. We apply a magnetic field of strength $B_0 \sim 92$ G with two permanent magnets, placed outside the cryostat chamber. From previous studies, it is already known that the quantization axis of the silicon vacancy centre on a cubic lattice site (usually referred as V2 centre)[28,29] is parallel to the defect centre studied in this report[31]. Therefore, we take advantage of an ensemble of V2 centres found at the edge of the solid immersion lens, which was likely created by ion bombardment during the FIB milling. For a perfectly aligned magnetic field, the splitting between two outer spin resonances of the V2 centre is $4|D_{\mathrm{gs,V2}}| = 140.0$ MHz for $|B_0| > |2 \cdot D_{\mathrm{gs,V2}}|$[9,28]. Here, we measure a splitting of $139.93 \pm 0.04$ MHz, meaning that the magnetic field is aligned within ~1.3 degrees. From ground state spin spectra as in Fig. 2b, c at the aligned magnetic field, we determine the ZFS of the V1 centre ground state, $2 \cdot D_{\mathrm{gs}} = 4.5 \pm 0.1$ MHz.

## Data availability

All data are available upon request from the corresponding authors.

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

## Acknowledgements
We would like to warmly thank Sébastien Tanzilli, Sina Burk, Rainer Stöhr, Roman Kolesov, Philipp Neumann, Sebastian Zeiser, Matthias Pfender, Thomas Öckinghaus, Charles Babin, Durga Dasari, Ilja Gerhardt and Stephan Hirschmann for technical help and fruitful discussions. R.N. acknowledges support by the Carl-Zeiss-Stiftung. R.N., M.M., M.W., Y-C.C., F.K. and J.W. acknowledge support by the European Research Council (ERC) grant SMel, the European Commission Marie Curie ETN "QuSCo" (GA No 765267), the Max Planck Society, the Humboldt Foundation, and the German Science Foundation (SPP 1601). S-Y.L. acknowledges support by the KIST institutional program (2E29580), C.B. acknowledges support by the EPSRC (Grant No. EP/P019803/1), and the Royal Society. N.T.S. received support from the Swedish Research Council (VR 2016-04068 and VR 2016-05362), the Carl Trygger Stiftelse för Vetenskaplig Forskning (CTS 15:339), the Knut and Alice Wallenberg Foundation (KAW 2018.0071), and the Swedish Energy Agency (43611-1). T.O. acknowledges support from JSPS KAKENHI 17H01056; A.G. acknowledges the Hungarian NKFIH grants No. NN118161 of the EU QuantERA Nanospin project as well as the National Quantum Technology Program (Grant No. 2017-1.2.1-NKP-2017-00001). A.G. and J.W. acknowledge the EU-FET Flagship on Quantum Technologies through the project ASTERIQS. F.K. and J.W. acknowledge the EU-FET Flagship on Quantum Technologies through the project QIA.

## Author contributions
R.N., S-Y.L., F.K. and J.W. conceived and designed the experiment; R.N., and F.K. performed the experiment; R.N., P.U., C.B., J.M., O.S., A.G., S-Y.L., F.K., and J.W. analysed the data; J.U.H., R.K., I.G.I., and N.T.S. prepared and characterised materials; T.O. contributed to electron beam irradiation; R.N. fabricated solid immersion lenses; M.N. developed software for data acquisition and experimental control; M.N., M.W., and Y-C.C. provided experimental assistance; P.U., J.M., O.S., A.G. and J.W. provided theoretical support; R.N., N.T.S., O.S., A.G., S-Y.L., F.K., and J.W. discussed and wrote the paper. All authors provided helpful comments during the writing process.

## Additional information

**Competing interests:** The authors declare no competing interests.

