## [Peer Review File · Nature Communications]

Reviewers' comments:

Reviewer #1 (Remarks to the Author):

The authors, R. Nagy et al., report "High-fidelity spin and optical control of single silicon vacancy centers in silicon carbide". While I think, this paper provides clear signatures of controllable single spins using silicon vacancies, I find the results weak for a justified publication in Nature Communications for the following reasons. The authors may clarify and comment on the following points:

1. Silicon vacancies in Silicon carbide are well known as reported in particular NATURE | VOL 479 | 3 NOVEMBER 2011 and PHYSICAL REVIEW B 66, 235202 ~2002. The authors need to clearly state how their sample is prepared that makes it different from the ones reported in literature.
2. In particular, NATURE | VOL 479 | 3 NOVEMBER 2011 shows coherent control and measurement of coherence time. Therefore, there is no real novelty in showing spin control in SiC in this paper. However, they achieve better coherence time. What is the reason behind the improvement? What is the limiting coherence time in comparison to other solid-state defects such as Nitrogen vacancies in diamond?
3. The authors need to clearly state the charge state of the Silicon vacancies. Since the authors have already referred to PHYSICAL REVIEW B 96, 161114(R) (2017) that reports the presence single isolated vacancies, the electronic structure in addition to the hyperfine coupling strengths, clarification is necessary to explain how this paper is an advancement over the previously reported results given that coherent control is already reported in literature.
4. Why are the resonant absorption spectra of five defect centers different as shown in Fig 1f? Could the authors clarify the meaning of "overlapping lines" as mentioned in the figure caption and reason?

Because of the recent interest in using color defects for quantum computation and sensing, the results are important, but the novelty is not clear from the current version of the paper.

Reviewer #2 (Remarks to the Author):

R. Nagy and colleagues reported on spin and optical control of single silicon vacancy in 4H-SiC, using experimental techniques based on photoluminescence excitation (PLE) and theoretical calculations based on density functional theory. The study has been carefully performed and the authors presented very interesting results with a number of convincing data both experimentally and theoretically. Silicon vacancy center in SiC is one of the emerging solid-state qubits that would definitely play a crucial role in both fundamental quantum information science and practical quantum applications. Thus, I suggest that the manuscript is potentially suitable for publication in Nature Communications. There are, however, several concerns about the data and results, which are summarized as follows.

1. Although this study presents very interesting results, there was a similar study, which was already published in PRX (Ref. 14). The previous work considered related defect qubits (di-vacancy centers) in SiC that has no inversion symmetry as well and also presented an efficient spin-to-photon interface using basically the same experimental techniques. In the current manuscript, it's hard to find the unique feature of this study. I know that the defect center is different and the spin and optical properties are different. But, are there any other special and unique features of this study compared to the previously published results, to warrant publication in this high-profile

journal?

Other issues:

2. For the readout and initialization fidelity, the authors mainly focused on the initialization fidelity, but the readout fidelity is not discussed at the same level. Although they are closely related to each other, I would think that it will be useful to discuss the readout fidelity as well to balance the discussion.

3. The authors reported the Hahn-echo coherence time of T₂ to be 0.85 ms. Considering that the SiC host is isotopically purified, the coherence time is relatively short. The authors discussed the possible origin in the supplementary information. However, to better identify the origin of the decoherence source, the following information would be helpful.

- Magnetic field dependence: the authors used only one magnetic field, which is 92 G. Did the authors consider other magnetic fields? If so, please provide the data.
- The temperature dependence of T₂ could also provide lots of useful information. Did the authors measure T₂ as a function of temperature?
- What was the exact fitting form of the Hahn-echo decay? Was it simple exponential or extended exponential? It would be also worth reporting the decoherence data on the other four single defect centers if the authors have.

Other minor issues:

In Fig. 1(f), the authors showed the spectra of five different centers and the authors claimed that the peak separation of all defects is nearly identical. Please report the exact numbers to compare the peak separation. And also for some of the defects, e.g. the black and green spectra, there are some minor side peaks next to the main peaks. What are those?

The authors showed that the excitation linewidth of the optical transitions approaches 60 MHz, and this is only twice the FT limit. I am wondering if the authors have further insight into the potential origin of the small residual spectral diffusion.

Reviewer #1 (Remarks to the Author):

The authors, R. Nagy et al., report “High-fidelity spin and optical control of single silicon vacancy centers in silicon carbide”. While I think, this paper provides clear signatures of controllable single spins using silicon vacancies, I find the results weak for a justified publication in Nature Communications for the following reasons. The authors may clarify and comment on the following points:

Before going into a detailed response to the referee’s comments we would like to underline the key-advances and the background of our studies.

There is now a world-wide search for optimal hardware for quantum information processing. In our case, we address systems which qualify as hardware for quantum repeater applications. Those require the following ingredients:

- 1) Readout and addressing of **single** quantum system.
- 2) High emission rate (and probability) of transform-limited photons as well as large absorption cross section of those photons.
- 3) Long memory times (realised by spins).
- 4) Operation on clusters of **single** spins for processing quantum information.

To date, no quantum system has shown to be capable of fulfilling all criteria simultaneously. Certainly, the **ensemble study**, cited by the reviewer (Nature **479** 84 (2011)), is elegant, however, it clearly lacks of demonstrating all ingredients 1) through 4) on a single quantum system.

We understand that in the original manuscript we lacked to make this point clear. We have thus rewritten the abstract and made substantial upgrades on the manuscript, including the introductory part, to help readers understand the main messages of our manuscript, in other words, it should be clear now that we have provided quite a decisive step forward in achieving those goals 1.) through 4.).

Further questions and comments are addressed in point-to-point responses.

1. Silicon vacancies in Silicon carbide are well known as reported in particular NATURE | VOL 479 | 3 NOVEMBER 2011 and PHYSICAL REVIEW B 66, 235202 ~2002. The authors need to clearly state how their sample is prepared that makes it different from the ones reported in literature.

The cited references, Nature **479** 84 (2011), is about the coherent control and optical readout (both with low fidelity) of **divacancy centre ensembles** in silicon carbide. Note that, apart from sharing the same host crystal, the divacancy and silicon vacancy centres have nothing else in common. The other reference is indeed a seminal work for the silicon vacancies, however, those **ensemble-based** measurements were intended to identify the basic electron spin properties of this defect and they are not at all related to optical spin control and readout. As both references provided by the reviewer do not present coherent spin control on the single defect level, they are markedly distinct from our work.

A potentially more relevant work is found in Nature Materials **14** 164 (2015), where in which coherent control of single spins in silicon carbide was demonstrated. Since then, we have made substantial advancements, such that we now demonstrate near-unity fidelity spin state initialisation, near-perfect

quantum state manipulation, longer spin coherence time, and optical properties that are (at least) on-par with the leading competitor (silicon vacancies in diamond). Note that all those properties are fundamental for quantum networking, and our work **shows for the first time** that those requirements can be met by colour centres in silicon carbide.

Regarding sample preparation, we agree with the reviewer that having a high quality sample is key. Major advances in this field have been made in the last couple of years and those advances are actually well-documented in the literature. In addition, we have extensively outlined sample preparation and characterisation methods in the Supplementary Information. Nevertheless, we have detail now even further some key aspects, e.g. how we isolated single silicon vacancies in which resonant optical transitions show very narrow linewidths, which is rather uncommon in solids. We back up our explanations by now adding additional experimental results on crystals with varying spin-defect densities. We benchmark our main sample in which the electron irradiation dose was 10^{12} cm⁻² to another one, where the dose was 10^{13} cm⁻². The higher irradiation dose leads to the formation of additional defects in the surrounding of the silicon vacancies. Spin and charge-state fluctuations of those additional defects are expected to lead to broader optical linewidths of the single silicon vacancies. Indeed, we observe ~150 MHz linewidths, which are ~2.5 times larger than in the low dose irradiated sample. This additional data tells that the low impurity concentration in combination with the unique property of the studied silicon vacancy (small Stark shift coefficient) leads to narrow (and stable) optical transitions.

Related with this remark, we added sentences “In previous works, we demonstrated that the V_{Si} shows low geometrical relaxation under excitation, leading to very high emission into the zero phonon line²⁶. Additionally, we performed preliminary investigations on coherent spin control and optical readout with limited contrast due to the lack of high-fidelity spin initialisation and readout procedures^{26,27}. Compared to our previous studies, the robust properties of V_{Si} in the improved host crystal material allows us to observe for the first time narrow and separated optical transitions of a single V_{Si} , which we address to demonstrate a high-fidelity spin-optical interface, suitable for quantum networking applications. Our findings prove that good quantum systems do not necessarily require inversion symmetry, which opens the door for new classes of quantum systems in numerous semiconductors and insulators.” in the introductory paragraph to introduce our previous works and provide contrast with respect to the submitted work.

We also added sentences after “Considering the 5.5 ns excited state lifetime²⁶, this is only twice the Fourier-transform limit, and might be explained by small residual spectral diffusion.” as below,

“For example, previous studies on NV centers in diamond and divacancies in SiC emphasize the impact of the sample quality, especially the impurity concentration, on the linewidth of the resonant optical transitions^{14,32}. We indeed find that a higher defect concentration leads to broader linewidths (see Supplementary Information). This suggests that low-damage defect generation techniques and Fermi level tuning via adapted doping can further enhance the quality of our optical transitions. We also measure the Stark shift tuning coefficient, which is at least one order of magnitude smaller compared to NV centres in diamond (see Supplementary Information). As explained in the Supplementary Information, this excellent optical stability is intimately linked to the wavefunction symmetry of the V_{Si} . Interestingly, unlike diamond, SiC cannot host defects with inversion symmetry due to non-centrosymmetry. As a consequence, the V_{Si} shows non-zero ground and excited state dipole moments,

which directly couple to electric fields. However, for the V_{Si} , those dipole moments are nearly identical, such that the optical transition energy is not altered by electric fields. Note that this does not preclude the existence of a strong dipole transition between ground and excited states, but it restricts the orientation to the symmetry axis of the defect, which is the c-axis of the crystal³³.”

2. In particular, NATURE | VOL 479 | 3 NOVEMBER 2011 shows coherent control and measurement of coherence time. Therefore, there is no real novelty in showing spin control in SiC in this paper. However, they achieve better coherence time. What is the reason behind the improvement? What is the limiting coherence time in comparison to other solid-state defects such as Nitrogen vacancies in diamond?

We stress again that divacancies and silicon vacancies share little, except the same host crystal. In addition, the cited work is based on spin manipulation of **ensembles**, while our work demonstrates high-fidelity initialisation and control of **single defects**. Since our initial work in Nature Materials **14** 164 (2015) and Phys. Rev. Appl. **9**, 034022 (2018), we have made substantial advancements, such that we now demonstrate near-unit fidelity spin state initialisation, near-perfect quantum state manipulation, coherence times long enough to reveal coherent coupling to nearby nuclear spins, and optical properties that are (at least) on-par with the leading competitor (silicon vacancies in diamond).

We indeed report the better coherence time, $T_2=0.85$ ms, compared to ~ 160 μ s and 84 μ s in our previous works, (Nature Materials **14** 164 (2015) and Phys. Rev. Appl. **9**, 034022 (2018)). Our decoherence test reveals the lower concentration of paramagnetic impurities which plays an important role for the better decoherence time. Please note that if the electron spin bath is diluted further, the coherence time will be limited by the nuclear spin bath. Our host crystal was enriched with two spin-less isotopes, ^{28}Si and ^{12}C , which provide a diluted nuclear spin bath. However, as we explained in the Supplementary Information S6, we created silicon vacancies at the cut-surface at the sample edge at which many undesired defects are present. We used this surface for aligning the electric field of the incident laser light to the crystallographic c-axis which is parallel to the optical axis of the V1 centre in 4H-SiC. In future studies, we will circumvent this problem by improved polishing or utilizing the a-plane substrate in which the c-axis lies in-plane of the sample. Then, we expect to observe even further improved spin coherence time.

Our results show that the long coherence times are mainly attributed a very weak spin-phonon coupling. The weak spin-phonon coupling is actually a key advantage of our system compared to silicon vacancies in diamond, where decent spin properties are only achieved at sub-100 mK temperatures and in high-strain environment (where inversion symmetry might become compromised/lost).

Regarding the improvement of T_2 , we added after “As shown in Figure 3(d), we measure a spin coherence time of $T_2 = 0.85 \pm 0.12$ ms”, an additional explanation as below,

“, which is longer than T_2 of a single V_2 centre, a V_{Si} at another non-equivalent lattice site, in a commercially available SiC wafer measured at room temperature²⁷. This improvement is thanks to the lower impurity concentration in the used CVD grown SiC layer (see Methods and Supplementary Information).”

To emphasize the benefit from the improved decoherence time, we mention now the ~ 1 kHz resolution of the quantum probe in the revised abstract. Additionally, after the “The Fourier transformation reveals

strong frequency components at $\omega_\alpha/2\pi = 77.9 \pm 0.1$ kHz and $\omega_\beta/2\pi = 76.0 \pm 0.1$ kHz.”, we added “This small difference (1.9 kHz) is resolved thanks to the high sensitivity of the quantum probe, the electronic spin with a nearly millisecond-long coherence time.”.

The limit of the decoherence time in the natural silicon carbide was experimentally and theoretically studied as in Simin, et al. Phys. Rev. B **95**, 161201 (2017) and Yang, et al. Phys. Rev. B **90**, 241203 (2014). Thanks to the hetero-nuclear spin bath, in which strong mismatch between ^{29}Si and ^{13}C Zeeman levels exist, one expects to longer decoherence time if the electronic spin bath is substantially diluted or its decoherence dynamics is suppressed by dynamical decoupling. However, the limit in the isotopically purified sample has not been studied yet to our best knowledge. Even though this is very important aspect in this research field, we do not want to provide furthermore discussion than what we already provided in the Supplementary Information of the submitted manuscript, since this is beyond of the scope of the current work.

3. The authors need to clearly state the charge state of the Silicon vacancies. Since the authors have already referred to PHYSICAL REVIEW B 96, 161114(R) (2017) that reports the presence single isolated vacancies, the electronic structure in addition to the hyperfine coupling strengths, clarification is necessary to explain how this paper is an advancement over the previously reported results given that coherent control is already reported in literature.

The cited paper (co-authored by 3 PIs of our team), presents a theoretical study on the electronic structure of the silicon vacancy in 4H-SiC backed up by EPR ensemble data. We do agree with the reviewer that we missed to state that the investigated centre is negatively charged. To help readers find this information more easily, we added “and negatively charged” after “As shown in Figure 1(a), we investigate the defect centre that is formed by a missing silicon atom at a hexagonal lattice site” and mentioned it also in the revised abstract.

Coming back to the key advancements of our work compared to previous studies, as outlined in the previous answers to the reviewer:

Compared to previous work on silicon vacancy centres in silicon carbide, we demonstrate near-unit fidelity spin state initialisation, near-perfect quantum state manipulation, coherence times long enough to reveal coherent coupling to nearby nuclear spins, and optical properties that are (at least) on-par with the leading competitor (silicon vacancies in diamond). We achieve all properties in the same system at the same time, which allows us to envisage the V1 centre in terms of a coherent spin-to-photon interface, and not just as a stationary qubit. To deliver this message more clearly, we now provide a new abstract and many parts in the introductory part have been added and/or modified (see the reply to the 1st comment) such as “Among colour centres in SiC, an efficient spin-to-photon interface has been demonstrated by divacancy spins in SiC¹⁴, however, the zero-phonon line (ZPL) of divacancies is notoriously sensitive to stray electric fields similarly to that of NV centre in diamond²⁵.”, and “Here, we show an alternative pathway to solving the above-mentioned issues, i.e. we resort to systems with highly symmetric ground and excited state symmetries and little redistribution of electron density between the two states. The resulting little change in electric dipole moments between the two states effectively decouples the system from electric field fluctuations. Interestingly, strong optical transitions can still be obtained, provided that

ground and excited states wavefunctions have alternating phases. In this report, we show that the hexagonal lattice site silicon vacancy (V_{Si}) in the 4H polytype SiC close to ideally matches all criteria.”.

Indeed our observation of a **single** weakly couple nuclear spin is an interesting and substantial step towards the use of this type of defect for quantum networks. It shows that our electron spin coherence times are good enough to observe coupling to nuclear spins even at nanometre distances. We added “**We emphasize that we did not observe hyperfine coupling in Hahn echo measurements on the other defects, however an adapted crystal growth with nuclear spins at a percentage concentration should allow to identify multi-spin clusters that could be used as small-scale quantum computers⁴³.**”.

4. Why are the resonant absorption spectra of five defect centers different as shown in Fig 1f? Could the authors clarify the meaning of “overlapping lines” as mentioned in the figure caption and reason?

The observed five spectra show almost identical separation between two resonant lines and linewidths. They look different since the centre of the line distribution is inhomogeneously distributed over a few 100 MHz. Many other defect centres show such inhomogeneous distributions due to coupling of the wavefunction to external electric fields (see for example Phys. Rev. Lett. **97** 083002 (2006)). In our studies, line positions did not shift with and without electric fields, so that we attribute local strain to be responsible for the observed shifts. We created solid immersion lenses around the V1 centres, which leads to inhomogeneous strain, depending on the exact position under the solid immersion lens. A more systematic study of the line shift via externally applied strain would certainly be interesting, however it is beyond the scope of the presented work. To explain it, we added “**The observed inhomogeneity is likely due to local strain near the surface of the solid immersion lens which is fabricated to improve the photon collection efficiency.**”, after “...allowing us to identify several defects with overlapping emission”

In Figure 1(f), some of the spectra show overlapping absorption lines. For example, defects #2 and #4, as well as defects #3 and #5. We mentioned this overlapping since many high-level quantum networking experiments are crucially based on optical interference, which requires identical/indistinguishable photon emission of multiple quantum systems (see for example Nature **558** 268 (2018)). To avoid confusion of non-experienced readers, we have added “**almost identical separation between two ZPLs and linewidth but inhomogeneously distributed.**” in the caption.

Because of the recent interest in using color defects for quantum computation and sensing, the results are important, but the novelty is not clear from the current version of the paper.

We thank this referee for viewing our results to be important. The main point of this paper is the demonstration of an efficient spin-to-photon interface with high fidelity that is robust against stray electric fields without bearing inversion symmetry, in a technologically mature material. Our finding has a great importance in the field in terms of searching for other colour centres in non-centrosymmetric materials. We have made several improvements to our manuscript to clarify the immediate impact that our results have in the field and we believe that the novelty of our work is now clearly stated.

Reviewer #2 (Remarks to the Author):

R. Nagy and colleagues reported on spin and optical control of single silicon vacancy in 4H-SiC, using experimental techniques based on photoluminescence excitation (PLE) and theoretical calculations based on density functional theory. The study has been carefully performed and the authors presented very interesting results with a number of convincing data both experimentally and theoretically. Silicon vacancy center in SiC is one of the emerging solid-state qubits that would definitely play a crucial role in both fundamental quantum information science and practical quantum applications. Thus, I suggest that the manuscript is potentially suitable for publication in Nature Communications. There are, however, several concerns about the data and results, which are summarized as follows.

We thank this Reviewer for suggesting that our manuscript is relevant and suitable for publication. By reading this Reviewer's comments and the ones of Reviewer #1, we realised that we did not put a strong enough emphasis on the key results and implications in the first version of our manuscript. We accordingly revised the manuscript, and backed up with additional data presented in the Supplementary Information. We now give a point-to-point response to the second reviewer.

1. Although this study presents very interesting results, there was a similar study, which was already published in PRX (Ref. 14). The previous work considered related defect qubits (di-vacancy centers) in SiC that has no inversion symmetry as well and also presented an efficient spin-to-photon interface using basically the same experimental techniques. In the current manuscript, it's hard to find the unique feature of this study. I know that the defect center is different and the spin and optical properties are different. But, are there any other special and unique features of this study compared to the previously published results, to warrant publication in this high-profile journal?

The reviewer is right in that the so far studied single quantum systems in silicon carbide are the divacancy and silicon vacancy centres. It seems therefore natural to compare those systems. Besides sharing the same host crystal, there are however many profound differences between both systems. The divacancy centre is much like the NV centre in diamond, with all its advantages (good spin properties that are relevant for sensing applications) and disadvantages (limited number of coherent photons which limits the potential for scalable quantum networking applications and the highly sensitive ZPL to the electric field).

In our work, we demonstrate that the silicon vacancy centre is the very first one to combine millisecond spin coherence times with rock-stable optical properties at convenient temperatures of 4-5 Kelvin. In addition, we demonstrate a deeper insight into the underlying physics of "good" solid state quantum systems in view of quantum networking. Up to now, inversion symmetry was considered to be the key for achieving stable and narrow optical transitions, as demonstrated with SiV centres in diamond, and similar derivatives such as the recently investigated SnV, GeV, PbV centres in diamond.

In stark contrast, our system, the silicon vacancy centre is hosted in silicon carbide, cannot provide inversion symmetry. Yet, we find optical stability, which is on-par with the leading competitor, SiV in diamond. Therefore, our finding will trigger searching for other defects without inversion symmetry as

prospect candidates as stated in the introductory part, “By demonstrating that inversion symmetry is not a stringent criterion for an ideal quantum emitter, we open the door for a new class of quantum systems in numerous semiconductors and insulators.” which is revised to “Our findings prove that good quantum systems do not necessarily require inversion symmetry, which opens the door for new classes of quantum systems in numerous semiconductors and insulators.”. In this regard, we suggest that our work should not be compared to the work in ref. 14.

In more detail, the studied silicon vacancy features a high emission ratio into the ZPL (>40%), as we recently reported in Phys. Rev. Applied 9 034022 (2018). In contrast, the divacancy in silicon carbide and NV centre in diamond feature only 7% and 3-4%, respectively. Our system has therefore a significantly higher probability of generating spin-photon entanglement in emission. In addition, in the current manuscript, we report the narrow linewidth of the ZPL. The narrowest linewidth observed in our manuscript is 60 MHz, while 80 MHz was reported for the divacancy in Ref. 14. Indeed, these two values are seemingly similar. However, we should not compare the absolute values since the indistinguishability is determined by the relative ratio with respect to the lifetime-limited width. Since the lifetime is 5.5 ns and 14 ns for the V1 centre and the divacancy in 4H-SiC, respectively, the V1 centre’s linewidth is much closer to the lifetime-limit. This contrast becomes even more significant if we consider the remaining impurity concentration in the host crystal. In Ref. 14, the impurity concentration was very low, $5 \times 10^{13} \text{ cm}^{-3}$, while in our studies, we have at least one order of magnitude higher concentration of $6 \times 10^{14} \text{ cm}^{-3}$ (as outlined in the S6 of the Supplementary Information). It is also known that the ZPL of divacancies is sensitive to the stray electric fields, similarly to NV centre in diamond [APL 111 26 26403 (2017)]. As explained in the main text, we provide theoretical calculations to prove that the high-quality optical properties are obtained thanks to the unique insensitivity of the silicon vacancy centre, coming from its unique electronic structure. The almost identical dipole moment in the ground and excited state results in small macroscopic electric dipole moment as explained in S6 of the Supplementary Information. Our statements are backed up by our Stark shift measurements, which show no line wandering even at very high electric fields.

In order to make this clear, we modified and added new sentences into the introductory part and re-arranged the 4th and 5th paragraph of the main text, and added more explanations (see the replies to other reviewer’s comments). “Considering that a large fraction of the photons are emitted in the zero phonon line (>40%), this implies that spin-photon entanglement can be generated at rates of several tens to hundreds of Hz¹⁸, currently mainly limited by the non-optimised light collection efficiency.” is added in the last paragraph. A very recent reference, <https://arxiv.org/abs/1811.02037>, is also added to “Note that this does not preclude the existence of a strong dipole transition between ground and excited states, but it restricts the orientation to the symmetry axis of the defect, which is the c-axis of the crystal”. Further key assets of our system are the reported 97 % spin initialization fidelity, which is the best value among the colour centres in SiC, including ref. 14. We added “and mark the state-of-art value for colour centres in” in the last sentence of the second last paragraph. We also added “The optical resonance excitation allows for high-fidelity spin-selective initialisation which result in 97% optical contrast of spin readout.” in the last paragraph and “The maximum contrast, $1-I_{\max}/I_{\min}$ is $97 \pm 1\%$ ” in the caption for figure 3b to place more stress on the high-fidelity initialization and advanced readout techniques which result in the high readout contrast as well. In addition, our latest studies show already coherent coupling to nearby nuclear spins, which have immediate potential for quantum memory

applications and multi-qubit entanglement. To the best of our knowledge, such a demonstration is still lacking for the divacancy in silicon carbide.

Other issues:

2. For the readout and initialization fidelity, the authors mainly focused on the initialization fidelity, but the readout fidelity is not discussed at the same level. Although they are closely related to each other, I would think that it will be useful to discuss the readout fidelity as well to balance the discussion.

It is true that the readout fidelity is important as well. However, we believe that a similar study as was done with NV centres in diamond [Nature 477, 574–578 (2011)] cannot be provided at this stage. As outlined in the manuscript, all optical transitions are spin-conserving, such that a good readout fidelity is potentially achievable. However, all excited states show also a fairly strong coupling to a metastable state manifold, from which non-selective decays back to the ground state occur. This means that optically-assisted spin state readout is somewhat inefficient. In other words, provided that a photon is detected during the resonant readout, then the spin state of the system is inferred with near-unity fidelity. However, if no photon is detected, no statement can be made about the system's spin state. We have actually mentioned several pathways to obtain near-unity efficiency (and fidelity) for the readout, which we are about to investigate. For example, the system's second excited state, namely the V1' manifold is efficiently decoupled from the intersystem-crossing (Phys. Rev. Appl. 9, 34022 (2018) and Phys. Rev. B 93, 081207 (2016)) as stated in the last paragraph, and might provide an efficient readout. Regarding this comment, in order to deliver our message clearly, we added **“and single-shot readout of electronic spins, similar to protocols based on cold atoms or ions⁵⁰.”** in the second last sentence. Another pathway that we are also now investigate is a nuclear-spin assisted high-efficiency single-shot readout, in a similar fashion as experiments that have been carried with NV centres in diamond. The idea is to repeatedly map the nuclear spin onto the electron spin (CNOT gate) and perform the (low-efficiency) electron spin readout until a statistically relevant sample is acquired. To stress this point, we also add a new sentence **“The high optical contrast of ground spin state readout will also allow for single-shot readout of coupled nuclear spins.”** into the last paragraph.

3. The authors reported the Hahn-echo coherence time of T₂ to be 0.85 ms. Considering that the SiC host is isotopically purified, the coherence time is relatively short. The authors discussed the possible origin in the supplementary information. However, to better identify the origin of the decoherence source, the following information would be helpful.

- Magnetic field dependence: the authors used only one magnetic field, which is 92 G. Did the authors considered other magnetic fields? If so, please provide the data.

- The temperature dependence of T₂ could also provide lots of useful information. Did the authors measure T₂ as a function of temperature?

As explained in Ref.16 of the Supplementary Information, we expect a higher impurity concentration near to the surface compared to the bulk. As the investigated defects are all located close to the surface

at which the crystal dicing has been performed, we expect also high densities of larger defect clusters. Therefore, we attribute the source of decoherence to electron spins from other defects surrounding the tested silicon vacancy. Since one of the intrinsic defects, carbon vacancies are easily created and cannot be removed by the annealing temperature that we used, carbon vacancies are very likely the dominant source for the measured decoherence rate. Although, we agree that more systematic studies using many silicon vacancies will be helpful to identify the exact origin. However, it requires a substantial amount of experiments such as the temperature and magnetic field dependencies [Nat. Commun. 7, 12935 (2016)] and various dynamical decoupling techniques [Science 330, 60–63 (2010)] which will be subject of future work. Furthermore, since the main scope of the submitted work is the high fidelity control and initialization of the silicon vacancy spins using the spin-selective optical transition and the test of it for the use of spin-to-photon interface, we think such decoherence studies should be dealt in a separate paper.

Regarding this issue, we added a sentence at the end of the paragraph in which the decoherence is discussed in the main text,

“Unambiguous identification of the decoherence source will require systematic investigation by controlling spin bath dynamics [Science 330, 60 (2010), Nat. Commun. 7, 12935 (2016)].”

- What was the exact fitting form of the Hahn-echo decay? Was it simple exponential or extended exponential? It would be also worth reporting the decoherence data on the other four single defect centers if the authors have.

The Hahn-echo decay was fitted using an extended exponential function $f(t) = A \cdot \exp(- (t/T_2)^n) + y_0$. The least-square error fit to the data resulted in an exponent $n=3.45$. Hahn-echo measurements on other defects resulted in coherence times T_2 ranging from 0.80 ms to 0.88 ms.

Other minor issues:

In Fig. 1(f), the authors showed the spectra of five different centers and the authors claimed that the peak separation of all defects is nearly identical. Please report the exact numbers to compare the peak separation.

We thank the reviewer for pointing out this missing information. We have updated the manuscript accordingly and mention now the distribution of the zero-field splitting in the main text. **“As shown in Figure 1(f), the peak separation of all defects is nearly identical within 19 MHz.”**

And also for some of the defects, e.g. the black and green spectra, there are some minor side peaks next to the main peaks. What are those?

It is true that some small peaks are observable. They can have physical origin, but we also cannot exclude the possibility that a spectrum from an adjacent silicon vacancy which is captured in a definite confocal volume. Even though it is difficult to investigate their properties due to their small intensities, we plan to investigate for identification of these peaks in near future.

We added a sentence,

“We also can observe small additional peaks in a few defects, whose origin is not yet investigated.”.

The authors showed that the excitation linewidth of the optical transitions approaches 60 MHz, and this is only twice the FT limit. I am wondering if the authors have further insight into the potential origin of the small residual spectral diffusion.

Although we show that the silicon vacancy in silicon carbide is robust against spectral diffusion thanks to its well-overlapping ground and excited state wavefunctions, some residual spectral diffusion remains (see Fig.1a in the main text and Table S1&2 in the Supplementary Information). Table S1 shows significantly smaller but not zero coupling coefficient between the electric fields and the ZPL for V1 center w.r.t. NV center in diamond (Table S2).

To illustrate this, we show below the square moduli of the wavefunctions (electron density multiplied with the sign of the wavefunction; red (blue) is positive (negative) isosurface) of an ideal, hypothetical defect quantum emitter with no spectral diffusion and that of V1 center in 4H SiC. V1 center is close to but not exactly ideal quantum emitter so residual spectral diffusion is expected to the observed.

As discussed in S5 of the Supplementary Information, there are many other defects such as carbon vacancies, carbon antisite-vacancy pairs, carbon interstitial clusters, shallow nitrogen impurities. They can be ionized by optical illumination, thus resulting in injecting free carriers, which can form fluctuating electric fields. Non-zero coupling, then, will cause residual spectral diffusion. Although this explanation sounds reasonable, because we have not investigated the ionization of donors in surrounding of SiC under the given experimental conditions, we think it is speculation. Thus, without a quantitative analysis of possible field fluctuations, we believe that it is better to provide further explanations in the future when we will have additional samples at hand.

REVIEWERS' COMMENTS:

Reviewer #2 (Remarks to the Author):

I carefully read the rebuttal letter and the corrected manuscript. I found that the authors successfully addressed all the issues raised by the reviewers. Thus, I am happy to fully recommend its publication in Nature Communications.

Reviewer #3 (Remarks to the Author):

The authors report "High-fidelity spin and optical control of single silicon vacancy centers in silicon carbide". SiC is considered to be one of the most promising materials for quantum information processing, particularly for "scalable quantum network". So far, the available systems suffer from too large electron phonon interaction, resulting in low transform limited photon emission, or fast spin dephasing, requiring cooling to mK temperature. However, authors demonstrate that the negatively charged silicon vacancy centre in silicon carbide does not suffer from both drawbacks.

I read the revised manuscript, all comments of reviewers, and the replies of the authors. The reviewers read it in detail and gave many helpful comments, so most of my questions and comments were included in them.

My comments which were not mentioned by the reviewers are as follows.

The authors have raised the advantages of single silicon vacancy centers compared with colors centers (NV centers, SiV centers) in diamond as hardware for quantum repeater applications. In addition, in my opinion, one of another advantage of SiC (as hardware for quantum repeater applications) compared with diamond is a recent development of nano-fabrication techniques, such as a photonic crystal. To appeal the advantage of SiC (as hardware for quantum repeater applications), I suggest to mention it and describe the perspective toward "scalable quantum network".

The following paper is the recent research of it as an example.

B. S. Song, S. Jeon, H. Kim, D. D. Kang, T. Asano, S. Noda, "High-Q-factor nanobeam photonic crystal cavities in bulk silicon carbide" Applied Physics letters 113, 231106 (2018).

I think that the authors replied well to the comments from the reviewers and the revised manuscript is well written. Thus, I believe that the manuscript is suitable for publication in Nature Communications.

Replies to the reviewer's comments:

Reviewer #2 (Remarks to the Author):

I carefully read the rebuttal letter and the corrected manuscript. I found that the authors successfully addressed all the issues raised by the reviewers. Thus, I am happy to fully recommend its publication in Nature Communications.

We appreciate the reviewer's recommendation.

Reviewer #3 (Remarks to the Author):

The authors report “High-fidelity spin and optical control of single silicon vacancy centers in silicon carbide”. SiC is considered to be one of the most promising materials for quantum information processing, particularly for “scalable quantum network”. So far, the available systems suffer from too large electron phonon interaction, resulting in low transform limited photon emission, or fast spin dephasing, requiring cooling to mK temperature. However, authors demonstrate that the negatively charged silicon vacancy centre in silicon carbide does not suffer from both drawbacks.

We appreciate this reviewer's accurate understanding of our works.

I read the revised manuscript, all comments of reviewers, and the replies of the authors. The reviewers read it in detail and gave many helpful comments, so most of my questions and comments were included in them.

My comments which were not mentioned by the reviewers are as follows.

The authors have raised the advantages of single silicon vacancy centers compared with colors centers (NV centers, SiV centers) in diamond as hardware for quantum repeater applications. In addition, in my opinion, one of another advantage of SiC (as hardware for quantum repeater applications) compared with diamond is a recent development of nano-fabrication techniques, such as a photonic crystal. To appeal the advantage of SiC (as hardware for quantum repeater applications), I suggest to mention it and describe the perspective toward “scalable quantum network”.

The following paper is the recent research of it as an example.

B. S. Song, S. Jeon, H. Kim, D. D. Kang, T. Asano, S. Noda, “High-Q-factor nanobeam photonic crystal cavities in bulk silicon carbide” Applied Physics letters 113, 231106 (2018).

We agree with this reviewer's perspective. We added this reference in the Discussion section as ref. [47].

I think that the authors replied well to the comments from the reviewers and the revised manuscript is well written. Thus, I believe that the manuscript is suitable for publication in Nature Communications.

We appreciate the reviewer's recommendation.